# Key Challenges to Optimal Therapeutic Coverage and Maternal Utilization of CMAM Program in Rural Southern Pakistan: A Qualitative Exploratory Study

**DOI:** 10.3390/nu14132612

**Published:** 2022-06-24

**Authors:** Farooq Ahmed, Najma Iqbal Malik, Nudra Malik, Madeeha Gohar Qureshi, Muhammad Shahzad, Muhammad Shahid, Sidra Zia, Kun Tang

**Affiliations:** 1Vanke School of Public Health, Tsinghua University, Beijing 100029, China; jam007@uw.edu (F.A.); de202159006@uibe.edu.cn (M.S.); 2Department of Anthropology, Quaid-i-Azam University, Islamabad 44000, Pakistan; 3Department of Psychology, University of Sargodha, Sargodha 40100, Pakistan; najma.iqbal@uos.edu.pk; 4Department of Applied Psychology, Lahore College for Women University, Lahore 54000, Pakistan; nudra.malik@lcwu.edu.pk; 5Pakistan Institute of Development Economics, Islamabad 44000, Pakistan; madeeha.qureshi@pide.org.pk; 6Institute of Social and Cultural Studies (ISCS), Bahauddin Zakariya University, Multan 60800, Pakistan; mshahzad@bzu.edu.pk; 7School of Insurance and Economics, University of International Business and Economics (UIBE), Beijing 100029, China

**Keywords:** CMAM, therapeutic barriers, disparities, social inclusion, South-Punjab, Pakistan

## Abstract

Severe Acute Malnutrition (SAM) is a serious public health problem in many low- and middle-income countries (LMICs). Therapeutic programs are often considered the most effective solution to this problem. However, multiple social and structural factors challenge the social inclusion, sustainability, and effectiveness of such programs. In this article, we aim to explore how poor and remote households face structural inequities and social exclusion in accessing nutrition-specific programs in Pakistan. The study specifically highlights significant reasons for the low coverage of the Community Management of Acute Malnutrition (CMAM) program in one of the most marginalized districts of south Punjab. Qualitative data are collected using in-depth interviews and FGDs with mothers and health and nutrition officials. The study reveals that mothers’ access to the program is restricted by multiple structural, logistical, social, and behavioral causes. At the district level, certain populations are served, while illiterate, and poor mothers with lower cultural capital from rural and remote areas are neglected. The lack of funding for nutrition causes the deprioritization of nutrition by the health bureaucracy. The subsequent work burden on Lady Health Workers (LHWs) and the lack of proper training of field staff impact the screening of SAM cases. Moreover, medical corruption in the distribution of therapeutic food, long distances, traveling or staying difficulties, the lack of social capital, and the stigmatization of mothers are other prominent difficulties. The study concludes that nutrition governance in Pakistan must address these critical challenges so that optimal therapeutic coverage can be achieved.

## 1. Introduction

Worldwide, nearly 20 million children suffer because of SAM. Every year, half of the global childhood mortality is caused by malnutrition and one-third of these deaths are caused by SAM alone [1]. South Asia and Sub-Saharan Africa show the highest rates of underweight and stunting [2,3]. Almost 78% of wasted children belong to the three South-Asian nations: Pakistan, Bangladesh, and India [4]. As every sixth person in Pakistan lives in poverty [5,6], the rate of child malnutrition in the country is higher than those of other South-Asian nations [7]. 

Previous evidence around the globe shows that supplementary programs have reduced moderate and severe acute malnutrition in children [8,9,10,11]. Recently, governments across the world adopted multisectoral strategies to address the problem of malnutrition. These strategies combine nutrition-specific and nutrition-sensitive indicators. However, evidence showed that the multisectoral solution strategy remained less successful in achieving the desired results [12,13].

In Pakistan, a nutrition-specific CMAM therapeutic program was set up in Southern Punjab’s poverty-stricken and flood-affected districts to deal with SAM. Under the CMAM program, moderate as well as noncomplicated SAM cases are treated with Ready-to-Use Therapeutic Food (RUTF), whereas complicated SAM children are referred first to the nutrition Stabilization Center (SC) by LHWs. Once complicated SAM cases become stabilized by using specialized medical milk 75/100, they can use RUTF. Ingredients in RUTF depend on the local acceptability availability and cost, but a standard RUTF is made up of milk powder, peanut butter, vegetable oil, vitamins, minerals, and sugar. The advantage of the product is its long shelf-life without refrigeration, but its demerit is its import as it is not prepared locally.

A sufferer’s experience is a social product shaped by structural violence, which may be defined as violence “built into the (social) structure and shows up as unequal power and consequently as unequal life chances” [14] (p. 171). Kawachi et al. [15] observed that unequal health hazards for individuals are the product of social, economic, cultural, and political processes in society because health outcomes are curtailed by the exploitative apparatuses of resource distribution, power, and social control. Structural violence is indirectly exercised by different parts of the social machinery of oppression and is apparently “nobody’s fault“ [16]. Similarly, Quesada, Hart, and Bourgois [17] (p. 339) defined structural vulnerability as “a product of class-based economic exploitation and cultural, gender/sexual, and racialized discrimination and processes of symbolic violence and subjectivity formation that have increasingly legitimized punitive neoliberal discourses of individual unworthiness”. 

State institutions and programs ignore certain individuals based on caste, gender, and class, thus subjecting them to indirect violence. This results often in the failure of development programs [18], and children and mothers with lower social and cultural capital bear the brunt of this structural violence [19]. As these development programs often fail to achieve their stipulated targets, a lasting impact of such programs is that the poor in the target population becomes indifferent to similar interventions in the future. They tend to deprioritize health and normalize disease and malnutrition [20] rather than seeking to benefit from government intervention due to their negative experience of these programs. The legacies of underdevelopment, stigma, and discrimination, along with insufficient public healthcare systems, lead to poorer health outcomes for rural poor and ethnically marginalized households.

State institutions, development, and poverty alleviation programs often ignore the individuals belonging to poor, rural, and lower castes [21]. Inequalities based upon castes, gender, and class in South Asia have failed development programs because they marginalized poor and weaker members [18,19], which resulted in maternal and child health disparities [19]. In South Asia, the poor often face difficulties becoming beneficiaries; therefore, evidence [22] suggested that area, gender, caste, and class determinants of social exclusion must be advised for program objectives, eligibility criteria of clients, and the selection process. Social capital is required to access medical settings [23]. Along with it, studies have explained that the corruption within the government’s medical settings in Pakistan and India showed a lot of parallels [24,25,26]. 

This study gives the narratives of healthcare providers and mothers of SAM children seeking treatment from the therapeutic program in the district of Rajanpur of Punjab province in Pakistan. This qualitative study contributes to the literature by describing barriers and resources while accessing nutrition-specific services. The study focuses on the issues of health sector corruption, structural inequalities, and the role of social capital. It adds to critical medical anthropology and the public health literature. It also investigates challenges and barriers to health and therapeutic coverage, why the government lacks interest in the implementation of the nutrition-specific program, and how the poor are generally secluded.

## 2. Materials and Methods

### 2.1. Data Collection

The qualitative data for this study were collected during fieldwork in the Rajanpur district of South Punjab from January to May 2017. This area was selected purposefully because it was flood-affected, poverty-stricken, and where female illiteracy and maternal-child malnutrition rates were highest in the whole province. Development infrastructure such as healthcare facilities was also scarce and where rural poor women face disparities.

This exploratory study was based on a purposive selection of key stakeholders involved in the CMAM program including healthcare providers and mothers of malnourished children (Table 1). After reviewing the available literature and using keywords such as social barriers and structural challenges to therapeutic coverage [10,27,28,29], a semi-structured interview guide was developed for interviewing, which was pre-tested with a few respondents and also updated from time to time whenever more information about the issue was revealed during the fieldwork. Exploratory research, as a methodological approach, investigates those research questions that have not previously been studied in depth. Exploratory research is often qualitative, involving a limited number of respondents, but is in-depth in nature. Therefore, only the most relevant stakeholders were interviewed: healthcare providers first (supply side), because they might have been cooperative in introducing other key stakeholders, i.e., mothers of malnourished children enrolled in the therapeutic program. Thus, in the next phase, mothers of malnourished children (demand side) were interviewed for this study. 

First, Key Informants Interviews (KIIs) with key officials of the District Health Authority were conducted face to face by the principal author who has experience in public health nutrition and knowledge of medical anthropology. Secondly, a Focus Group Discussion (FGD) with LHWs was conducted in a healthcare facility by the two qualitative researchers (F.A. and S.Z.). In the group discussion, 10 participants were maximally allowed to take part. Participants of this discussion were inquired about the major difficulties, barriers, and challenges that hampered therapeutic coverage at the district level. Finally, healthcare providers helped to identify and communicate with mothers having SAM children. The mothers of malnourished children were identified by the Nutrition Assistants appointed at SCs and LHWs involved in the CMAM program. To seek consent to take part in this research, 30 mothers were informed about the nature of the study. However, consent could be agreed upon by 20 mothers. We chose the respondents’ places deliberately so that they felt safe and comfortable. Audio recorders were not used, owing to locals’ comfortability and cultural sensitivity. In-Depth Interviews (IDI) were in a flexible format, ranging from one to two hours. All interviews were conducted face-to-face in the local language (Seraiki). The open-ended in-depth interviews continued until experiences and essences were repeated, and until information saturation was achieved through 10 mothers (Table 1). The majority of the mothers of SAM children were either uneducated or had a few years of schooling along with minimal socio-cultural capital and disadvantaged economic status (i.e., <USD 100/month). 

### 2.2. Data Analysis

Without delays, researchers translated verbatim all the qualitative data obtained from group discussions, semi-structured interviews, and field notes from the local language to the English language. Then, we reviewed all the raw data available and labeled all sentences and text into different colors and codes to find out the common meanings. After this, we grouped similar codes to create broader categories. Next, we had to cross-verify the narratives and remove the inconsistencies, vagueness, and discrepancies. Lastly, codes and categories were analyzed and different themes that affected therapeutic coverage were identified using inductive research methods. In total, seven prominent subthemes subsequently emerged from the whole exploratory qualitative data. In the end, all conspicuous challenges, barriers, and difficulties were subsequently assembled into five leading themes: (1) politico-economic or financial, (2) administrative and planning, (3) logistical, (4) social or cultural capital, and (5) behavioral or interactive (see Figure 1).

### 2.3. Ethical Considerations

The ethical approval for this study was acquired from the advanced study and research board (AS&RB) of Quaid-e-Azam University Islamabad in its 307th meeting held on 20 October 2016. The board committed to approving the endorsements of the Dean of the Faculty of Social Sciences to accept the current qualitative and ethnographic research in the Department of Anthropology. In addition to this, the Department of Health District Rajanpur also approved the study protocols and tools. All the participants were thoroughly informed about the nature and purpose of this study before taking their formal consent to be part of this exploratory qualitative research. As the majority of mothers were illiterate, oral consent was provided according to their wish and comfortability. After taking informed verbal consent from all study participants, we promised to ensure their anonymity, privacy, and confidentiality.

## 3. Results

Our overall qualitative findings revealed the emergence of multiple financial, administrative, logistical, and behavioral difficulties that challenged the CMAM therapeutic program for the treatment of severely malnourished children in the Southern Punjab region of Pakistan.

### 3.1. Financial Barriers

Health priorities at the micro-level are influenced by macro-level incentives. Funding for different national or provincial health or nutrition programs determines the focus of health staff.

#### 3.1.1. Funding and Priorities of Health Bureaucracy

The national Polio Eradication Program, being the most favorite program, was prioritized by the health bureaucracy. The health department used most of its energy in this program and deprioritized others. 

*“Although the nutrition program has been functional for many years, the staff isn’t free to run this at the district level. The health office gives importance to their routine matters and does not let this kind of vertical program be implemented in full scale and strength”.* (Nutrition Official, KII)

#### 3.1.2. Work Burden on LHWs

LHWs coordinate between the community and health department; therefore, they were involved in almost every program, whether provincial or national. They frequently complained that they faced extra work pressure and burden, particularly from the Polio eradication program. Their primary duty was to cover and coordinate with more than two thousand pregnant and lactating females in their concerned outreach areas. Over-involvement reduced their concentration in their original work about child and mother work. The over-involvement of LHWs de-prioritized nutrition activities by the health department. 

*“LHWs are involved in other programs, especially Polio. After working three to five days in the Polio campaign, one LHW would not go into the field because she is already tired. Similarly, in Measles, LHW is fully engaged for 12 days and becomes so fatigued and rarely visits the field for some days, and demands rest. When the department asks working overly and extensively, how she can fill the high gaps created in the nutrition program. This pressure is regular; Polio and other activities are unfinishable”.* (LHW, FGD)

*“Funding availability in the Polio eradication program was the leading cause of why the health department Punjab always engaged LHWs for only this at the stake of another important program because their funding was low or none. It was owing to this fact that LHWs always wandered for Polio drops and skipped nutritional screening and education”.* (LHW, FGD)

In Southern Punjab, several LHW posts were vacant according to reports of the district health information system. Out of a total of 900, only 650 LHW seats were filled, which showed that the covered-up population in the district was 44%. LHWs felt dissatisfaction with the lower salary packages and other allowances. Logistical and cultural hurdles, along with extra workload, jointly restricted their will and motivation. On many occasions, many of them took this duty as no more than a formality just because they could not merely refuse orders from the department. As a result, they ignored visiting assigned households regularly due to low salaries and poor economic incentives. Many LHWs were not well trained in anthropometric measurements of mothers and children for screening purposes. Unfortunately, these LHWs in the least developed areas were not appointed or even remained absent. Many of these LHWs reported that their performance was perfect, and they tried to justify their role. They always report that everything was going well. One official remarked:

*“LHWs are called almost every week, sometimes for meetings, sometimes for training, or sometimes for another task. She has to maintain and carry multiple registers. I mean, it’s a serious matter that needs to be seen and fixed. The patients from remote rural and tribal areas are missed; SAM cases are from remote areas, where there is a water problem, and access is limited. So cases mostly come from rural areas”.* (Health Official, KII)

### 3.2. Administrative and Planning Failures

The training of field staff, screening, referral of SAM cases, and distribution of therapeutic food are compromised due to the weak administration of the program.

#### 3.2.1. Improper Utilization of Nutrition Field Staff: Lack of Training

In 2008, the Government of Punjab recruited Health and Nutrition Supervisors at the BHU level to screen and train the community on common health diseases and nutritional issues. However, many of the remote BHUs missed them as there was no infrastructure. Since their creation, they have barely taken part in any significant nutrition intervention in the district. Their role in CMAM was never acknowledged until recently when the multisectoral nutrition center (MNSC) at the provincial level anticipated their future participation in the province of Punjab in a report in 2017. They were not fully trained on nutritional issues and, hence, lacked relevant knowledge about the causes and treatment of malnutrition. It was reported that an international organization (Micronutrient Initiatives (MI)) had trained them on the importance of micronutrient iodine for mothers and children. Therefore, these supervisors were mostly assigned monitoring duties for Polio, EPI, and dengue prevention programs instead of nutrition. However, they were properly trained on malnutrition for the first time in 2017 after 9 years of recruitment, which showed the lack of coordination and failure of the precise job description. This also showed a weak commitment to combatting malnutrition, lack of vision, and relevant policy failures. The staff appointed at remote health units rarely performed duties because of insecure environments, a lack of monitoring mechanisms, dilapidated hospital buildings, damaged roads, and a lack of transport facilities. These isolated areas are those where more attention is needed. Most recently, new district coordinators were recruited by the multisector nutrition center who also need training on nutrition issues.

*“There are gaps….as the district coordinator of the malnutrition addressing committee has only one or two meetings with the Deputy Commissioner of the district. Also, MSNC established by the Planning and Development Commission of Punjab province has recruited district coordinators, but they are new and have no significant work to do. Nutrition supervisors are also not so trained and involved, nor can they help measure and refer malnutrition cases, but their involvement is limited to the polio program. Although all these have been appointed, they have no work to do, except work on special weeks. Recently, we called nutrition supervisors on nutrition week. They were assigned to distribute multi-nutrient sachets in their schools as area in-charges, but they are not really in much coordination”.* (Health Official, KII)

#### 3.2.2. Weak Referral, Indifference, and Interpersonal Conflicts among Staff

Intrahospital or staff interpersonal politics at the Basic Health Units (BHUs) level emerged as one of the most significant reasons behind the weak referral of SAM cases to the SC at District Headquarters Hospital (DHQ). It was remarked that:

*“The cases which reach DHQ without a referral are admitted right away, but SAM referral is constrained and slow, especially, people from remote rural areas are in great need because of the weak and poor referral system to the Stabilization Centre. Every month LAMA (who quit treatment) cases are increasing; 4–5 SAM cases are admitted daily, totaling approximately 120–150 in one month. Most of these cases are located at the basic health unit (BHU) level. For the treatment of SAM, it is very difficult to screen a child with a complication from the field by these LHWs through Mid Upper Arm Circumference (MUAC). LHW refers these SAM cases to Lady Health Visitor (LHV) who has to verify MUAC and complications, and forward complicated SAM cases to DHQ by an “1134 ambulance service”.* (Nutrition Official, KII)

After the anthropometric screening, LHWs generally referred malnourished mothers and children to BHUs and Rural Health Centers. However, many mothers were kept waiting unnecessarily by Lady Health Supervisors (LHSs) appointed at BHUs. Many poor and illiterate mothers left health units because they felt they were being ignored, unattended, and devalued by these LHSs. 

*“LHW and LHV are often at odds with each other. Sometimes LHS dislikes an LHW, who insists on checking children immediately. Every LHW expects that she has hardly convinced and referred parents of SAM case to BHU, so now LHS should give priority so that it could be further referred to Stabilization Centre at DHQ. LHS asks LHW to ‘wait outside’ and does not attend to the case even after hours. This is how SAM cases leave hope for treatment and run away, and this is why referral of severely malnourished children with complications is minimum. However, a child specialist and nutrition staff, specified for this work only, are readily available at SC; therefore, SAM cases are measured and admitted without trouble. However, people from only nearby areas can reach directly to SC, but cases from remote areas have to be ignored”.* (Nutrition official, KII)

#### 3.2.3. Lack of Monitoring and Medical Corruption

The presence of formula milk companies inside hospitals and the sale of not-for-sale RUTF were two significant factors. Although banned theoretically, representatives of the multinational formula milk were reported to move freely in the SC, BHUs, and RHCs for advertising and selling formula milk to the poor parents of severely SAM children. 

*“Soon after recovering from complicated SAM, mothers were motivated to try their products. The company trains its agent to remain alert and keep an eye on every person monitoring and conducting research. They are well trained in rapport building with medical staff and patients’ attendants for convincing them to use their products after the advertisement. Nobody ever restricted such active advertisement and sale”.* (LHW, FGD)

Therapeutic food was reported to be sold out at the hands of some LHWs. These packets of therapeutic food are not for sale. It was informed by community members that the Plumpy-Nuts were being sold out at some places at the price of PKR 20–30 per sachet by a few LHWs. A mother indicated: 

*“I requested our LHW to give some food but she refused. I threatened one such LHW who used to sell it by saying, ‘give some sachets for my son, or else I would complain against you that you sell off the therapeutic food illegally.’ Never were any actions taken against such complaints by the concerned authorities”.* (Mother of SAM child, IDI)

*“The distribution of therapeutic food is not altogether transparent and fair. Health staff often prefer and prioritize their relatives and close ones first whenever the task of providing therapeutic food is given to them”.* (Mother, IDI)

### 3.3. Lack of Social and Cultural Capital among Poor Mothers

Relationships with those who control power, access to information, and interpersonal skills necessary to communicate are essential requirements for becoming beneficiaries of development programs.

#### Rural-Urban Disparities: Accessing Therapeutic Program

When asked whether the field staff visited your area or household and what were the impacts of therapeutic food, most respondents agreed that the milk provided at the stabilization center and RUTF had a good impact on the sick child. The majority of the enrolled mothers in CMAM showed that their children were recovering gradually. In their opinion, the specialized medical milk (75/100) and RUTF brought a positive impact on their severely malnourished children. When asked how mothers came to know about the treatment of severely malnourished children at stabilization centers or CMAM, most parents revealed that they were referred by the medical community or people from urban centers told them about this program and suggested visiting the nutrition Stabilization Centre at DHQ to obtain special milk (75/100) for malnourished babies. 

*“Doctors, LHWs, and active community members helped to refer us to the CMAM program and SC, for therapeutic 75 milk for the severely malnourished baby”.* (Mother closer to the city area, IDI)

*“LHWs visited our area and told us to bring milk from CMAM staff; vaccinators also visit and inform us about the program”.* (Mother from Peri-urban area, IDI)

*“LHWs do not visit our area, but vaccinators do once a year so we sometimes bring our children to the hospital for immunization and sometimes not. People from the city informed us about this program; they suggested us to visit Stabilization Centre because milk [75/100] was being distributed there”.* (Mother from the remote village, IDI)

Only a few parents reached the SC without any referral, which implies how different local forms of social capital or relationships helped mainly urban or peri-urban families to become beneficiaries and isolate and seclude the majority of the most deserving remote, rural, illiterate, and lower-income families with lower social capital. 

*“We, the females, are carrying this unfortunate child without any help from other family members. I am a mother, how can I leave him alone in this condition, only my heart knows how much disturbed I am. No one can realize the state of my heart; I cannot see my child suffer. I am in profound psychological distress. When will my child feel normal and healthy, I don’t know. I have tried my best to make him healthy and nourished. We have wandered everywhere, here and there, to find if someone could suggest a better way. Recently a person from our neighborhood informed us about this program, I requested my mother to test this place [Stabilization Center] too”.* (Mother from the remote village, IDI)

### 3.4. Logistical Difficulties

Treatment of complicated SAM children requires their mothers and other caregivers to stay at the SC for some days until the child is stabilized and can come to the simple RUTF stage. However, most mothers complained they had to leave medical advice due to logistical hurdles.

#### 3.4.1. Geographic Seclusion: Difficulties in Traveling

The poorest of the poor mostly live in risky, far remote, and underdeveloped areas. Geography is one of the central causes of inequities in health and nutrition. Results showed that distance emerged as a substantial barrier to coverage and access to health and nutrition programs. Logistical problems emerged as the most significant reasons for low access. The bad transportation, long travel times, damaged roads, and long distances to the site were the major determinants of little coverage. Females are less empowered in these settings due to the lowest access to healthcare facilities and literacy and employment opportunities. One mother stated, “We are tired and we still have to travel”. The mother informed that they reached the stabilization center after much difficulty and running errands:

*“The [Nutrition stabilization] center is very far from our village, and it took hours to get there. We had to catch several types of transport; the first motorcycle from our community to another town, then an auto-rickshaw to the main highway. After it, we had to catch a bus from the road to reach the district bus stand. From the bus stand to the hospital, we had to hire an auto again. After wandering here and there madly in the hospital building, we reached the stabilization center by asking for addresses with the help of so many people. We got tired when we arrived here, and we still have to travel, we’ll have to go back home as it is not allowed to stay without permission”.* (Mother of complicated SAM Child, IDI)

#### 3.4.2. Problems Related to Staying at the Stabilization Center

Many mothers insisted on the hospital staff that they wanted to treat the complicated SAM children at home. SC staff had objections to this idea because the condition of the severely malnourished children was unstable, and they were required to stay until the children became stable in the center. The Punjab government previously claimed to set up SCs at the sub-district level, but activities at the SCs were being limited at the district level, and at any time, the program may come to an end. UNICEF in Pakistan has recently intended to study the bottlenecks in the CMAM program. This indicates that these programs are still under the control of UN agencies and the government lacks ownership.

*“Convincing parents about the treatment at SC is a very complex task. Mental preparation of family and parents is essential for this because a mother or someone from the family has to stay for at least four days. They have to prepare their basket or bag”.* (LHW, FGD)

The other strong reason for low therapeutic coverage was the loss of income if the mother and father were to stay at the SC receiving the treatment for only one severely malnourished child and ignoring the rest of their children. This made them indifferent to complete treatment. Therefore, most grandmothers had to stay at the SC. Mothers could not stay longer, because no one could take care of the rest of the children at home. During crop season, poor rural mothers could rarely afford to give proper time for treatment and health-seeking. Some domestic servants also complained about working hours. As they could not escape from their duty, they delayed check-ups and treatments of complicated SAM children. If mothers had to stay, they had to bring all of their kids along with them to the SC at the district headquarters hospital. As children were unaware of cross-infections at the sites, they were playing in the hospital’s wards, touching the floor with their hands, and eating foods there without handwashing.

### 3.5. Behavioral Problems with Nutrition Staff

Another critical factor of low coverage of the therapeutic program in rural and Southern districts of Punjab province in Pakistan involves the elements of stigma, respect, and dignity.

#### 3.5.1. Stigmatization of Patients and Attendants

Many poor parents felt stigmatized and complained of being unattended at the hands of the hospital and nutrition staff. Illiterate people with low socioeconomic status had low confidence to communicate with hospital staff and feared being insulted by the doctor and staff. The behavior of the staff was not supportive. Sometimes staff felt irritated by the poor’s dirty clothes. CMAM staff was often reported to have been rude to mothers of severely malnourished children. 

Multiple times, mothers indicated taunts and offensive remarks and the mothers felt ashamed of this embarrassing situation. For example, on one occasion, a nutrition assistant at the SC vocalized to a mother, “*you are always here to get this milk*”. Once, a female nutrition staff member threw the packets of formula milk 75 toward a mother in a very disgusting mood and said angrily, “*hold this packet and get out*”. In another instance, when a poor mother brought her child to the stabilization center for the treatment of SAM, the on-duty staff responded “*take your dirty luggage from here; it smells stinky*”. 

#### 3.5.2. Not Being Attended

Complaints about not being attended to by low-income parents were much more common. Mothers explained how the staff at the nutrition stabilization center was indifferent, careless, and rude.

*“We would wait all day and night, but no person attended a little. The sick child used to cry all night as they would give our child nothing to eat and drink. We were worried when the doctor and staff would pay attention to our child. Leaving such treatment [of indifference and disgust] would be better than just wasting time [in wait] here”.* (Mother at SC at DHQ, IDI)

*“My husband said to SC staff ‘my child is hungry, and you pay no attention. I do not want to leave my sick child as hungry all night.’ Nurses complained about my husband to the head doctor, who called him and insulted him. My husband got disheartened and finally decided to quit the treatment at this center”.* (Mother at SC at DHQ, IDI)

## 4. Discussion

This study discovered mothers’ interactions with the biomedical treatment and therapeutic system of the CMAM program and nutrition stabilization center. It specifically explored how poor, illiterate, and rural women were often incapable of navigating the therapeutic coverage and politics with institutions. These difficulties were perilous for many women, mainly from remote and secluded areas, who were illiterate and lacked the required minimum cultural assets and social skills to negotiate the complex and unfamiliar setting [29]. Women’s communications with the health and nutrition staff illuminated how the administration strengthened health and nutrition inequities. Barriers related to geography, income, fears of maltreatment, and discrimination emerged as most striking and significant for the rural poor struggling to receive therapeutic care through the public healthcare system [20,30]. Many families could not access the CMAM program, owing to multiple socio-cultural and logistical reasons [27,28]. The staff of development programs often secluded poor mothers and children due to multiple power dynamics [22,23,31]. 

Families, who had some links within local power circles (social and cultural capital), received better chances of coverage. To combat the problem of malnutrition, the government needed to change priorities. At the primary level, deprioritization of the “nutrition program” in comparison with the “Polio eradication program” resulted because of heavy international funding for the latter. This suggests that the government must increase funds for nutrition [32]. Further, the burden and pressure on the LHWs must also be curtailed by focusing their attention on maternal-child health and nutrition programs. In remote areas, seats for the LHWs ought to be urgently allocated [33]. Nevertheless, all these steps require road construction and infrastructure provisions at basic health facilities. Human development infrastructure at the local level is also required in South Punjab, which is always facing regional or ethnic inequalities [34].

The poor rely on traditional treatment methods because of their low income. Stigmatization and the trust deficit of the poor in government departments are strong indicators of low biomedical service utilization along with expensive and uncontrolled private clinicians’ prevalence that need urgent policy decisions. This study showed that the medical staff did not care much about the marginalized victims of social stigma and ignored the poor’s feelings [35]. The future design and implementation of government programs must be made more socio-culturally sensitive. Plumpy nuts are effective only in emergency contexts but not in chronically poor settings, and such programs also create a dependency of low-income states on international companies, which prepares such foods. In addition, therapeutic food is not available in usual and regular circumstances, even though people need it. The permanent solution, therefore, lies not in treating the individual body but in searching for a cure for a social body through political-economic means of social justice and equity [36]. 

Evidence showed that nearly half of the population in several rural districts was not covered by LHW, especially in the most remote and the poorest areas [37,38]. UNICEF [39] has highlighted that the neonatal mortality rate was reduced in low-caste groups where LHWs made weekly visits in rural Indian Punjab. Recent studies [40,41] similarly demonstrated that sufficient training, financial compensation, and close supervision of community health workers are imperative for the successful delivery of SAM treatment along with the adequate quantity of ready-to-use therapeutic food. 

Some respondents revealed that therapeutic food was being sold off by LHWs, and representatives of formula milk producers were free to move into hospital settings. There is evidence that formula milk companies ignore the laws and continue marketing their products inappropriately [42]. The literature from Pakistan and India shows that corruption within medical settings restricts government services [26,43]. While drawing upon the anthropology of the state along with the perspective of structural violence, Gupta found that funds hardly reach their anticipated beneficiaries but mostly reach people with political acquaintances, cultural capital, and financial influence [44]. Inaccurate systems of information based on statistics, conflict, and wide-scale corruption in Indian bureaucracy systematically isolate and ignore the poor. Similarly, examining the “Government of Papers, in Pakistan”, Hull [45] analyzed how the bureaucratic processes and management of records crafted partnerships among people as the core apparatus and governing emblem of the official measurement of bureaucracy. For him, papers should be seen “as mediators that shape the significance of the linguistic signs inscribed on them” [45] (p. 13), which shows that postcolonial bureaucratic records are materialized under the colonial policy of keeping government and society isolated.

Many poor, illiterate, and rural mothers indicated that they faced rude behavior and stigma in medical settings. In a study in the Kenyan context, analogous shame, stigma, and discomfort at health clinics related to malnutrition and fear of mistreatment at the hands of the biomedical staff were noted as the most significant barriers to treatment for childhood acute malnutrition [46], which potentially constrained their access to the CMAM program. Chary et al. [20] argued that childhood diseases are treated incompletely because of the perception that the child is not being attended to. They linked the phenomenon of “not being attended” with healthcare inadequacies. 

Our findings showed that mothers faced logistical difficulties. Evidence in Guatemala similarly showed that poor women suffered from running errands [27]. Similar evidence showed that therapeutic programs in five African countries failed because of the low awareness about the program, long distances, the handling of rejection at sites [29,30,31], and the centralization of the program [47]. The study’s findings corroborate that distant communities remained potentially disadvantageous to be covered by therapeutic programs particularly for the treatment of complicated SAM because caregivers had to stay for many days at the therapeutic center [29], more often adjacent to the children’s hospital. Evidence [48] from the adjacent Sindh province of Pakistan also showed that remote areas were less exposed to the therapeutic program, and the common barriers included the low awareness of malnutrition and its services, the children’s disapproval of RUTF, long distances, and high opportunity cost. This study also found that remaining in the program until full recovery was difficult.

This article monitors mothers’ interactions while accessing the nutrition-specific CMAM program. In doing so, it proposes that a “politics of neglect” is at play in these programs in neglecting the social body and poorer sections of society in the program’s target areas. These interventions do not consider processes of power and exploitation and ignore the complex and unequal social relations. The narratives showed how the poor often faced structural inequities and social exclusion due to a lack of social or cultural capital. Evidence showed that the poorest of the poor and low-caste families with the lowest social capital in Punjab were excluded from the cash transfers program (nutrition-sensitive program) at the will of local political leaders [49]. Some of the literature found similar results that only people with approaches and links to local politicians could be successful in becoming beneficiaries of the income support program in Pakistan [50]. Families with lower socio-cultural capital suffered the most because of the lack of transparent and impartial social protection policies and social safety nets. The literature from other contexts on so-called bureaucratic hurdles has highlighted such misery of poor women facing structural inequalities and the indifference of bureaucracy toward the poor people who have no relationships with influential notables [51]. The lack of social and cultural capital deprives the poor of their due rights, despite deserving them. On the other hand, people with such capital were witnessed on several occasions, becoming beneficiaries even if they did not deserve it well.

According to Bourdieu [52], cultural capital plays a vital role in taking benefits from society. When they are guided to adopt specific procedures, illiterate mothers cannot remember the steps and names of officers. The poverty eradication or development programs preferably target the better-off, ignoring many poor of the poorest who have never been taken seriously by the bureaucratic structure of the development programs. When resources are limited, competition is high; therefore, the humanitarian apparatus has to be narrow in its scope, leaving many deserving and potential beneficiaries far behind [53]. In Pakistan, poverty is extensive. The poorest are deprived because they lack links and relationships with people in power. Pakistan is not a place where resources are equitably distributed and where the population is also under control. This bureaucratic structure does not let the poor and weaker enter their offices unless an officer, lawyer, politician, or any other notable accompanies them. The poor often endure social and structural difficulties in the process of being beneficiaries, so knowledge about social exclusion is fundamental to advise on program objectives, eligibility criteria of clients, and the selection process [53]. 

In addition, CMAM is a short-term curative measure, especially in emergency contexts. Not aligned well with local socio-cultural realities, the short-term global technical solution in the form of RUTFs and CMAM was implemented “under neoliberal governments and facilitated an increasingly inequitable economy with minimal state involvement in an increasingly individualistic social environment”. [54] (p. 16). However, the permanent, long-term, sustainable solution to maternal child undernutrition lies in females’ socio-economic emancipation, and their health or nutrition literacy [55,56,57,58,59]. In addition, the inclusion of training of medical staff on respectful care is imperative. 

## 5. Conclusions

The CMAM program in Southern Pakistan encounters multiple social, economic, and structural obstacles. First, funding in nutrition as compared with other programs deprioritizes officials’ interest in nutrition and involves LHWs in other multiple tasks that increase their work burden and divert their attention from maternal-child health and nutrition. In addition, the corruption in food distribution and the unethical sale of RUTF by LHWs are reported, which need strict monitoring and fair dispensation. The normalization of social exclusion has roots in politico-economic and structural inequalities. The study includes the following recommendations: prioritizing more funding for nutrition; proper training of field staff; improving screening skills and referral of SAM cases; providing traveling incentives to needy, illiterate, and rural mothers; devolving the child stabilization service at the micro (UC/BHU) level; distributing RUTF fairly by LHWs; treating parents politely. Finally, vacant LHWs’ seats in remote rural areas demand urgent allocation.

## Figures and Tables

**Figure 1 nutrients-14-02612-f001:**
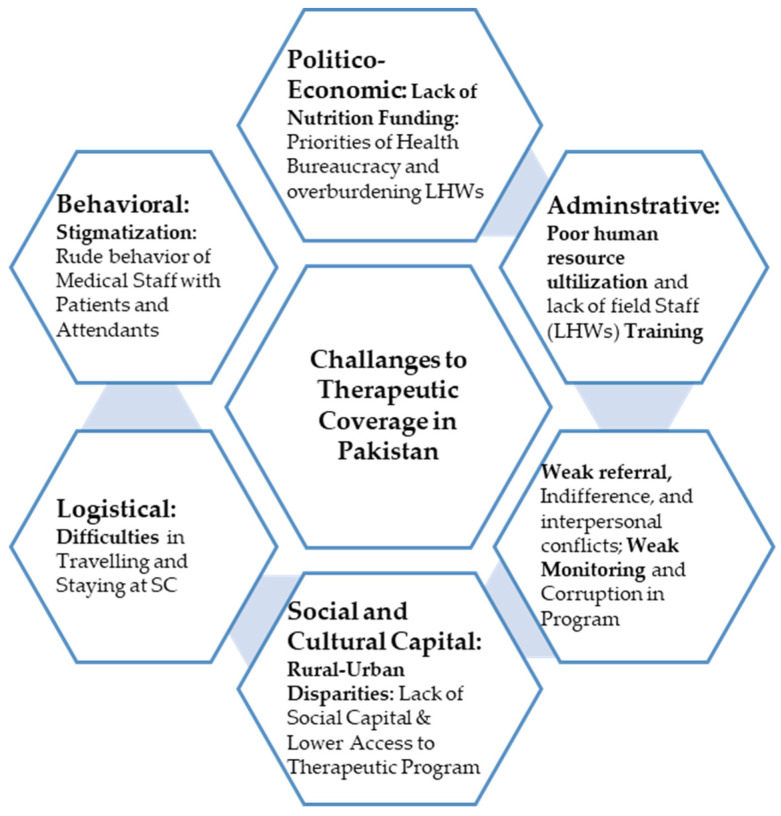
Graphical abstract of Significant Barriers to Optimal Therapeutic Coverage in Rural and Southern Pakistan.

**Table 1 nutrients-14-02612-t001:** Details of Interviews and FGDs with Study Respondents (*n* = 25).

**Description of Interviews and FGDs**	**No of Respondents (*n*)**
Key Informant Interviews (with Health Nutrition Officials)	5
In-depth Interviews (with Mothers of SAM Children)	10
FGD (with Lady Health Workers)	10
**Demographic and Social Characteristics (*n* = 25)**	**Frequency (Percentage)**
Gender	
Female	20(80%)
Male	5(20%)
Education	
Uneducated	7(28%)
5th to 8th Class	3(12%)
High School	15(60%)
Profession	
Agrarian Labor	5(20%)
Household Work	5(20%)
Salaried Class	15(60%)

## Data Availability

Not applicable.

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
