# Peer review of "Key Challenges to Optimal Therapeutic Coverage and Maternal Utilization of CMAM Program in Rural Southern Pakistan: A Qualitative Exploratory Study"

_nutrients, 2022, doi:10.3390/nu14132612_

Round 1
Reviewer 1 Report
Dear authors,
the topic you dealt with is interesting.
There are some points that need to be improved.
First of all, I recommend changing the title as given the low number of subjects interviewed in the three phases of the survey process it would be better to refer to a pilot survey or an exploratory survey. Another point that could represent a methodological weakness concerns points 2.3 Ethical Consideration. In fact, you refer to a protocol approved by the board of the faculty of social sciences. Even with the methodological rigor of approving the protocol, in order to avoid the risks of a self-certification of the model, it would be necessary to find scientific contributions that validate the process you have followed.
Another point to improve is to indicate the topics covered during the semi-structured interview supported by bibliographic references and also how the choice of samples took place during the three phases of the investigation process.
Author Response
The topic you dealt with is interesting. There are some points that need to be improved.
First of all, I recommend changing the title as given the low number of subjects interviewed in the three phases of the survey process it would be better to refer to a pilot survey or an exploratory survey.
Response: Thank you very much for this important suggestion. We have revised the topic as follows: (Please see Line 4)
Key Challenges to Optimal Therapeutic Coverage and Maternal Utilization of CMAM Program in Rural Southern Pakistan: A Qualitative Exploratory Study
Another point that could represent a methodological weakness concerns points 2.3 Ethical Consideration. In fact, you refer to a protocol approved by the board of the faculty of social sciences. Even with the methodological rigor of approving the protocol, in order to avoid the risks of a self-certification of the model, it would be necessary to find scientific contributions that validate the process you have followed.
Response: Thanks for this valuable comment. We revised the ethical statement in the 2.3 section and included the statement explaining that besides the department of Anthropology of the faculty of social science the study protocol and tools were also approved by the Health Department district Rajanpur, which is a medical institution.
The Department of Health District Rajanpur also approved the study protocols and tools. (Please see Line: 167-168)
Another point to improve is to indicate the topics covered during the semi-structured interview supported by bibliographic references. Also how the choice of samples took place during the three phases of the investigation process.
Response: Thank you for the important comments. We have rewritten and revised the statement (Please see Line 112-123)
After reviewing available literature and using keywords like social barriers and structural challenges to therapeutic coverage [10,27-29], a semi-structured interview guide was developed for interviewing which was also updated from time to time whenever more information about the issue was revealed during the fieldwork. Exploratory research, as a methodological approach, investigates those research questions which have not previously been studied in depth. Exploratory research is often qualitative involving a limited number of cases but is in-depth in nature. Therefore, only the most relevant stakeholders were interviewed, first healthcare providers (supply side) because they might have been cooperative in introducing other key stakeholders i.e. mothers of malnourished children enrolled in the therapeutic program. So, in the next phase, mothers of malnourished children (demand side) were interviewed for this study.
Reviewer 2 Report
In the present paper, Farooq Ahmed and colleagues aimed to explore how poor and remote households face structural inequities and social exclusion in accessing nutrition-specific programs in Pakistan. The study specifically highlights significant reasons for the low coverage of the Community Management of Acute Malnutrition (CMAM) program in one of the most marginalized districts of south Punjab. They showed that the CMAM program in Southern Pakistan encountered multiple social, economic, and structural obstacles. Moreover, the study concludes that nutrition governance in Pakistan must address these critical challenges so that optimal therapeutic coverage could be achieved.
Overall, I think that the paper could be of interest for readers and researchers, in general, on a relevant social issue. I make some suggestions for improve the quality of the manuscript.
1) Please better define and discuss the power analysis of study. In other words, the sample size is large enough, from a statistical point of view, to reach definitive conclusions?
2) The authors, if possible, should incorporate in tables the dietary pattern of the patients included in CMAM program (i. e. Mediterranean-style diet, Plants-based diet, Nordic dietary pattern, etc.). In this way, I feel that the readers can better understand the critical challenges that nutrition governance in Pakistan must address so that optimal therapeutic coverage could be achieved.
Author Response
In the present paper, Farooq Ahmed and colleagues aimed to explore how poor and remote households face structural inequities and social exclusion in accessing nutrition-specific programs in Pakistan. The study specifically highlights significant reasons for the low coverage of the Community Management of Acute Malnutrition (CMAM) program in one of the most marginalized districts of south Punjab. They showed that the CMAM program in Southern Pakistan encountered multiple social, economic, and structural obstacles. Moreover, the study concludes that nutrition governance in Pakistan must address these critical challenges so that optimal therapeutic coverage could be achieved.
Overall, I think that the paper could be of interest for readers and researchers, in general, on a relevant social issue. I make some suggestions for improve the quality of the manuscript.
1) Please better define and discuss the power analysis of study. In other words, the sample size is large enough, from a statistical point of view, to reach definitive conclusions?
Response: Thank you very much for this valuable point. In qualitative research, representation and generalization are not as important considerations as in quantitative research. Qualitative exploratory research, as a methodological approach, investigates those research questions which have not previously been studied in depth. Exploratory research is often qualitative involving a limited number of cases but is in-depth in nature. Therefore, only the most relevant stakeholders are interviewed.
2) The authors, if possible, should incorporate in tables the dietary pattern of the patients included in CMAM program (i. e. Mediterranean-style diet, Plants-based diet, Nordic dietary pattern, etc.). In this way, I feel that the readers can better understand the critical challenges that nutrition governance in Pakistan must address so that optimal therapeutic coverage could be achieved.
Response: Thanks for the important note. We have rewritten and revised statements.
CMAM is a therapeutic program for moderately and severely acute malnourished children in which the Ready-to-Use-Therapeutic-food is given to non-complicated cases. For complicated SAM children, specialized medical milk 75/100 is given for a few days at the stabilization center until complicated cases are made stable to use RUTF. Ingredients in RUTF depend on local acceptability availability, and cost but a standard RUTF has a composition of milk powder, sugar, peanut butter, vegetable oil, vitamins, and minerals. The shelf life of RUTF a long and require no refrigeration. In Pakistan, RUTFs are basically imported and not prepared locally.
We have explained the above discussion in the introduction section of the article:
In Pakistan, nutrition-specific CMAM therapeutic program was set up in Southern Punjab’s poverty-stricken and flood-affected districts to deal with SAM. Under the CMAM program, moderate as well as non-complicated SAM cases are treated with Ready to Use Therapeutic Food (RUTF) whereas complicated SAM children are referred first to the nutrition Stabilization Center (SC) by LHWs. Once complicated SAM cases become stabilized by using specialized medical milk 75/100 they can use RUTF. Ingredients in RUTF depend on local acceptability availability, and cost but a standard RUTF is made up of milk powder, peanut butter, vegetable oil, vitamins, minerals, and sugar. The advantage of the product is its long shelf-life without refrigeration but de-merit is its import as it is not prepared locally. (Line: 51-60)
Reviewer 3 Report
A good study, provides insight into important inequality and barriers to CMAM coverage in southern Pakistan. The study stimulates new important questions of how the poor/needy can be supported to access CMAM without discrimination and intimidation.
Minor changes to consider:
Abstract-Suggest LHW (24) be written in full first time.
Introduction:
1) Sentences 49 - 54 sound more of discussion of finding, suggest relocating to discussion section
2) 'Medical corruption' (92) is not clear, may need to be redefined/rephrased as the method does not mention any tool used to measure medical corruption
Methodology
106 - 110. If the qualitative tools were pretested, would be more informative to mention.
Conclusions
In the recommendation, 539-540, Suggest inclusion of training on respectful care.
Author Response
A good study, provides insight into important inequality and barriers to CMAM coverage in southern Pakistan. The study stimulates new important questions of how the poor/needy can be supported to access CMAM without discrimination and intimidation.
Minor changes to consider:
Abstract-Suggest LHW (24) be written in full first time.
Response: Thank you very much for this pointing out this mistake. We incorporated the full word Lady Health Workers before giving the abbreviation LHWs. (Please see Line: 29)
Introduction:
1) Sentences 49 - 54 sound more of discussion of finding, suggest relocating to discussion section
Response: Thank you very much for this point. The following paragraph has been relocated to the discussion section. (Please see Lines: 519-525)
This article monitors mothers’ interactions while accessing the nutrition-specific CMAM program. In doing so, it proposes that a ‘politics of neglect’ is at play in these programs in neglecting the social body and poorer sections of the society in the program's target areas. These interventions do not consider processes of power and exploitation and ignore the complex and unequal social relations. These narratives show how the poor often face structural inequities and social exclusion due to a lack of social or cultural capital.
2) 'Medical corruption' (92) is not clear, may need to be redefined/rephrased as the method does not mention any tool used to measure medical corruption.
Response: Thank you very much for this comment. We have replaced the word “medical corruption” with “health sector corruption.” (Please see Line 97-98)
Methodology
106 - 110. If the qualitative tools were pretested, would be more informative to mention.
Response: Thank you very much for this concern. After reviewing literature using keywords related to the challenges of therapeutic coverage, a semi-structured interview guide was developed for interviewing, which was pilot tested before data collection. It was also updated from time to time whenever more information about the issue was revealed during the fieldwork. (Please see Lines: 112-116)
In the recommendation, 539-540, suggest inclusion of training on respectful care.
Response: Thanks for the input. We have revised the sentence and incorporated the feedback. (Please see lines: 559-560)
However, the permanent, long-term, sustainable solution to maternal child undernutrition lies in females’ socio-economic emancipation, and their health or nutrition literacy [55-59]. In addition, the inclusion of medical staff training on respectful care is imperative.
Round 2
Reviewer 1 Report
Dear Authors,
the paper is improving. In my opinion can be published
Reviewer 2 Report
The authors have satisfactorily responded to all my questions and made the necessary changes to the manuscript.